# ACTIVATION FUNCTION INFORMED QUANTIZATION

## ABSTRACT

Large batch sizes in transformer-based language and vision AI applications mean that performance is increasingly bottlenecked by linear layer computation, and weight-only quantization only exacerbates this computational bottleneck. While full 4-bit weight and activation post-training quantization with no model quality loss remains an open challenge, we offer a novel approach of selective channel-wise quantized W4A4 and W8A8 computation. We observe that the gradients of transformer activation functions (ReLU, GELU, SiLU) are small when inputs are negative, which means that quantization error in pre-activation function inputs result in small output error. Exploiting this insight, we propose Activation Function Informed Quantization (AFIQ), which samples dot-product partial products on a single calibration example to determine which channels to quantize for all future model inference. We implement a mixed-precision linear layer kernel in CUDA to evaluate latency and we find that AFIQ linear layers are 17% faster than baseline with negligible loss in model quality.

## 1 INTRODUCTION

Transformer-based Vaswani et al. (2017) vision and language models are behind rapidly-growing and popular AI applications Altman & Anderson (2025), driving enormous demand for new compute infrastructure Huang (2025) and new model training Buchholz (2024). Making efficient use of compute infrastructure and pre-trained models during deployment is key to meeting the demand for AI applications at the lowest cost, with inference consuming a large proportion of data center power Wu et al. (2022). Improving the efficiency of transformer models would also make possible more AI applications in resource-constrained environments, such as on smartphones and on autonomous drones.

While the memory bottleneck has been the focus of prior quantization works Frantar et al. (2022); Lin et al. (2024), He et al. (2025) have found that there is a compute bottleneck in the linear layers at batch sizes of 32 or above. We find that prior works' focus on 4-bit weight quantization Frantar et al. (2022); Lin et al. (2024) exacerbates the compute bottleneck because they propose dequantizing the weights and computing in FP16. Analysis of this compute bottleneck can be found in Appendix A.

In this work, we address the compute bottleneck by selectively computing some of the dot products in the linear layer (Fig. 2) with 4-bit weights and 4-bit activations (W4A4) for vision transformers and W8A8 for LLMs. We select dot products to quantize based on our observation about the activation functions following linear layers (ReLU Nair & Hinton (2010), GELU Hendrycks & Gimpel (2016), and SiLU Hendrycks & Gimpel (2016)). The magnitudes of activation function gradients are smaller for negative pre-activation inputs as shown in Fig. 1 (top). Thus, error in the linear layer dot products that produce those negative pre-activation inputs have less impact on the output after the activation function. We find that 80% of pre-activation inputs are negative as shown in Fig. 1 (bottom), representing an opportunity to apply quantized computation to a majority of the computation in the linear layer. Analysis of why pre-activation inputs are mostly negative can be found in Appendix D.

We propose a post-training quantization technique called Activation Function Informed Quantization (AFIQ) that exploits the smaller gradient magnitude for negative inputs to activation functions to compute select channels of the linear layer in W4A4. Channel selection can be calibrated using a single sample of model input and is based on sampling dot product partial products to identify channels with a high proportion of negative pre-activation inputs. Channel-wise selection is done

to allow for practical implementation on current hardware, which we demonstrate by implementing a mixed precision CUDA kernel. Overall, we can achieve 17% reduction in latency in linear layer computations with negligible effects on model quality.

We make the following contributions:

- We observe that activation functions in transformers (ReLU, GELU and SiLU) have small gradient magnitudes when pre-activation inputs are negative
- We propose a post-training quantization (PTQ) technique called Activation Function Informed Quantization (AFIQ) that samples dot products of a single model input sample to identify channels on which to use W4A4 computation for ViTs and W8A8 computation for LLMs
- Implement a mixed FP16 and INT4/INT8 CUDA kernel to demonstrate speed up of the mixed precision approach when compared to all FP16
- Evaluate AFIQ on a selection of vision transformers (ViTs) and large language models (LLMs) resulting in 17% improvement in throughput, with small effects on model quality when compared to full precision ViTs and AWQ Lin et al. (2024) weight quantized LLMs

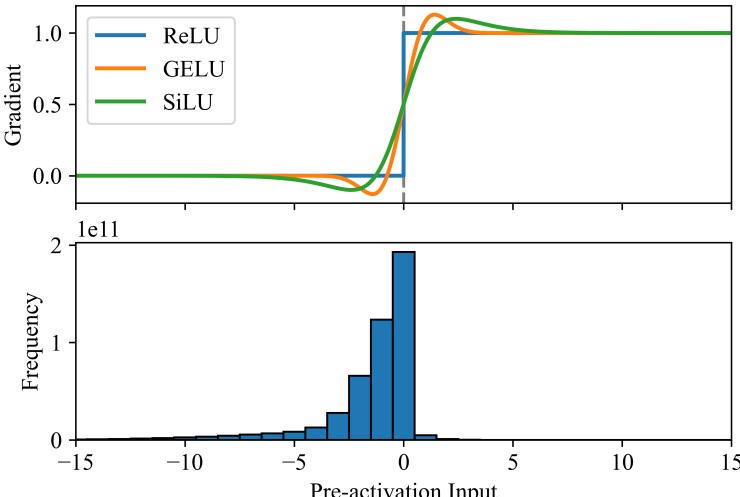

Figure 1: (top) The gradients of common transformer activation functions are plotted against pre-activation input values. Note that when the input is negative, the gradients of activation functions have smaller magnitudes. (bottom) Histogram of pre-activation function input values (from all layers in SwinTransformer Liu et al. (2021) on ImageNet-1K Deng et al. (2009)) shows left skewness, indicating a large proportion of negative values.

## 2 RELATED WORK

### 2.1 POST-TRAINING QUANTIZATION

Model quantization converts floating point weights and activations to lower precision formats, which can reduce memory costs by sending fewer bits and it can reduce compute latency by exploiting faster hardware. However, this can come at the cost of quantization error affecting model quality. While quantization-aware training Esser et al. (2019); Choi et al. (2018); Bengio et al. (2013) is a promising approach to producing quantized models, it involves resource-intensive training on a training dataset that may not be practical. Post-training quantization (PTQ), the approach chosen for this work, requires tuning a small number quantization parameters with a calibration dataset, which leads to minimal overhead for deployment. While PTQ efforts have also been directed at various other applications like diffusion models Li et al. (2023); Shang et al. (2023), LLMs and vision

transformers are the focus of this work. LLM.int8() Dettmers et al. (2022), PTQ4ViT Yuan et al. (2022), and SmoothQuant Xiao et al. (2023), which propose 8-bit weights and activations, allow for lower precision matrix multiplication, but not as low precision as the 4-bit computation proposed in this work. While we propose using W8A8 computation for LLMs like SmoothQuant Xiao et al. (2023), we use INT4 weights, which would alleviate memory bottlenecks. GPTQ Frantar et al. (2022), AWQ Lin et al. (2024), and SqueezeLLM Kim et al. (2023) propose quantizing weights to 4-bits or fewer, but require dequantization to FP16 for computation. While this addresses the *memory* bottleneck, it neglects the *compute* bottleneck problem addressed in this work. RPTQ Yuan et al. (2023) proposes W4A4 quantization with grouping channels into clusters with similar dynamic ranges, but suffers from some model quality loss. Indeed, it is the dynamic range of activations addressed in RPTQ that is preventing our work from quantizing to W4A4 for LLMs. Since AFIQ leverages a different insight of activation function gradient, we believe that RTPQ would be complementary to AFIQ.

## 2.2 Hardware Approaches

Prior works have used hardware approaches to exploit characteristics of activation functions, primarily ReLU. Some works Akhlaghi et al. (2018); Kim et al. (2021) propose ReLU-based early negative termination, which computes approximations of the pre-activation until the hardware can predict whether the pre-activation input is positive or negative. If the pre-activation input is negative, then computation can be terminated early, since all negative inputs to ReLU result in 0. These hardware proposals do not lend themselves to practical implementation on current GPUs, but they serve as inspiration for this work. For a deeper discussion of how these approaches might apply to transformers, which do not exclusively use ReLU, refer to Appendix C.

## 2.3 Quantized LLM Inference Software Frameworks

While LLM inference frameworks like TensorRT-LLM NVIDIA (2025b) and DeepSpeed Rasley et al. (2020) have made deploying compressed LLMs much easier, they do not support low precision computation, e.g. W4A4 matrix multiplication. On the other hand, COMET Liu et al. (2025) does propose using INT4 computation and describes their approach to switch between W4A4 and W4A8 depending on the presence of outliers in activation channels. Their selective quantization by *activation* channel is different from our approach of selective quantization by *weight* channel. Efficiently integrating activation channel and weight channel selective quantization would be interesting future work.

## 3 Preliminaries

### 3.1 The Transformer Architecture

Fig. 2 illustrates the modern transformer architecture, which is based on Vaswani et al. (2017). MLP blocks and multi-head attention blocks alternate with a residual connection bypassing each block. In the MLP block, there are typically two fully-connected layers with an activation layer in between. In this work, we are interested in the FC1 layer which computes the pre-activation inputs to the activation function. Note that for models that use SwiGLU Shazeer (2020), the FC1 layer is effectively split into gate projection and up projection components. For models that use SwiGLU, the pre-activation input refers to the output of the gate projection linear layer.

### 3.2 Activation Functions

The activation function between the fully-connected layers in the MLP varies from model to model with the most common being ReLU, GELU, SiLU Hendrycks & Gimpel (2016), and SwiGLU Shazeer (2020). GELU is used by DeiT Touvron et al. (2021), SwinTransformer Liu et al. (2021), and GPT Radford et al. (2018); ReLU is used by OPT Zhang et al. (2022); and SwiGLU is used by LLaMA Touvron et al. (2023a) and Qwen Team (2024). Additional analysis of these activation functions can be found in Appendix B.

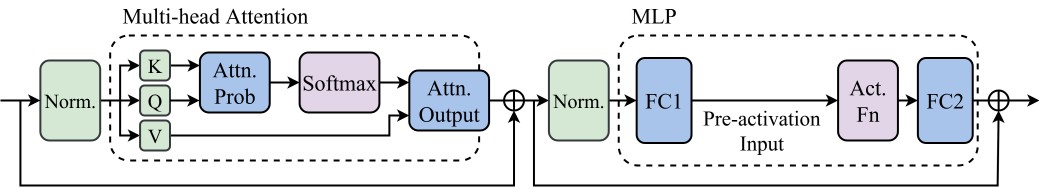

Figure 2: The transformer model architecture typically has alternating multi-head attention and MLP layers, with normalization layers and residual connections interspersed between them. This figure shows a basic block similar to the original transformer Vaswani et al. (2017), which is repeated to create a deep transformer network.

### 3.3 QUANTIZATION

Quantization reduces the number of bits required to represent a given value, by applying a mapping function $f$ to map the high precision value, $x_{\text{HP}}$, to the range [-1,1], multiplying by the maximum quantized representation for a given $N$ bits and rounding the result:

$$x_{\text{INT}} = \text{round}\left(\left(2^{N-1} - 1\right) f(x_{\text{HP}})\right) \tag{1}$$

To dequantize back to the original format, the reverse operation occurs:

$$x_{\text{HP, dequantized}} = f^{-1}\left(\frac{x_{\text{INT}}}{2^{N-1} - 1}\right) \tag{2}$$

Quantization introduces quantization error through the rounding process, but computation with quantized values can have higher throughput. On an A100 GPU, the peak INT4 throughput (FLOPS) is $4\times$ that of FP16 NVIDIA (2020). Quantized computations also use less energy Horowitz (2014).

## 4 METHOD

In this section, we describe how Activation Function Informed Quantization (AFIQ) improves computation throughput. First, we describe the method at a high level (§4.1), along with oracle experiments to demonstrate the potential (§4.2). Then, we analyze the gradient of the activation function to define the gradient prediction challenge (§4.3), one of several practical implementation challenges that we address. Specifically, our solutions include the First-K Sampling Predictor (§4.4.1), cached channel selection (§4.4.3), and mixed precision inference (§4.4.4).

### 4.1 OVERVIEW

AFIQ extracts the throughput improvements of quantized computation §3.3, while minimizing quantization error by avoiding quantization for positive pre-activation inputs. Consider a dot product in FC1 between activation vector $\mathbf{x}$ and weight vector $\mathbf{w}$, $\mathbf{x}' = \mathbf{x} \cdot \mathbf{w}^T$. If $\mathbf{x}$ and $\mathbf{w}$ were quantized, then their quantization errors would propagate to $\mathbf{x}'$. We would like to selectively quantize $\mathbf{x}$ and $\mathbf{w}$ for negative pre-activation inputs, since the gradient is smallest (Fig. 1). Since most pre-activation inputs are negative (80% in Fig. 1), there is the potential for most dot products to be quantized. However, current hardware support constraints means that AFIQ chooses channels that have a large proportion of negative pre-activation inputs and quantizes those channels.

### 4.2 ORACLE EXPERIMENT

Since the gradient is smaller when a pre-activation input is negative, negative pre-activation inputs should be more tolerant to quantization error. To test this hypothesis, we perform quantized computation of the FC1 layer (Fig. 2) of DeiT-Tiny Touvron et al. (2021) by quantizing the weight

and activations of the FC1 layer to 4-bits in channel-wise fashion using simulated quantization from Xiao et al. (2023). Then, we test four experimental settings to the input to the GELU: the baseline $f_1(x) = x$, fully quantized $f_2(x) = x_q$, and the following piecewise functions.

$$f_3(x) = \begin{cases} x, & \text{if } x > 0 \\ x_q, & \text{if } x \leq 0 \end{cases} \qquad\qquad f_4(x) = \begin{cases} x_q, & \text{if } x > 0 \\ x, & \text{if } x \leq 0 \end{cases}$$

where $x$ is the full precision FC1 result and $x_q$ is the quantized FC1 result, on an element-wise basis.

Our hypothesis would suggest that $f_3$ will perform closest to baseline since it uses the $x_q$ where the gradient magnitude is smallest. We note the confounding factor that 84.6% of pre-nonlinearity inputs are negative, so $f_3$ results in about $5.7\times$ the rate of quantization when compared to $f_4$. Nonetheless, we proceed with our experiment and the results are shown in Table 1.

Table 1: Oracle quantization experiment results

| GELU Input | Top-1 Accuracy |
|---|---|
| $f_1$ | 72.1% |
| $f_2$ | 65.7% |
| $f_3$ | 70.7% |
| $f_4$ | 68.0% |

The oracle experiment shows the potential for quantizing 84.6% of the computation with a 1.4% drop in accuracy when quantizing negative pre-activation input computation, compared to a 4.1% accuracy drop when quantizing the other 15.4% positive pre-activation input computation. The practical implementation (§4.4) will have the effect of reducing the proportion of computation quantized while also reducing the impact on model quality. An oracle study across all models can be found in App. G.

### 4.3 Gradient of the Activation Function

Consider the innermost loop of the FC layer and activation function:

$$y = \mathrm{F}_{\mathrm{act}}(\mathbf{x} \cdot \mathbf{w}^T + b) \tag{3}$$

where $\mathbf{w}$ is the weight vector, $b$ is the bias term, $\mathbf{x}$ is the input vector, and $\mathrm{F}_{\mathrm{act}}$ is the activation function. For convenience, define the pre-activation input $z = \mathbf{x} \cdot \mathbf{w}^T + b$, so $y = \mathrm{F}_{\mathrm{act}}(z)$.

The activation function gradient is input dependent, that is $\frac{\partial \mathrm{F}_{\mathrm{act}}}{\partial z}$ is a function of the pre-activation input $z$ (full derivation of this can be found in Appendix B). Additionally, because layer gradient depends on the activation function gradient, as shown below, the layer gradient also depends on the pre-activation input.

$$\frac{\partial y}{\partial x_i} = \frac{\partial \mathrm{F}_{\mathrm{act}}}{\partial z} \frac{\partial z}{\partial x_i} = \frac{\partial \mathrm{F}_{\mathrm{act}}}{\partial z} w_i \tag{4}$$

$$\frac{\partial y}{\partial w_i} = \frac{\partial \mathrm{F}_{\mathrm{act}}}{\partial z} \frac{\partial z}{\partial w_i} = \frac{\partial \mathrm{F}_{\mathrm{act}}}{\partial z} x_i \tag{5}$$

The dependence of the activation function gradient on the pre-activation input presents a circular dependency problem. While we would like to directly compute the activation function gradient to decide how to quantize FC1 computation, that requires having computed FC1 to produce the pre-activation input. To resolve this dilemma, we propose a lightweight predictor.

### 4.4 Practical Implementation

Expanding on the circular dependency problem discussed previously, the following are the practical implementation issues to resolve:

1. We do not know the sign of FC1 *a priori*, so we need to predict the sign to decide how to quantize

2. Sorting computation into full precision and quantized streams at an element-wise granularity would lead to unacceptable overheads

3. Our goal is to target hardware supported low-precision computation to achieve tangible benefits

### 4.4.1 FIRST-K SAMPLING PREDICTOR (FKSP)

To address challenge (1), we need a predictor for the sign of pre-activation inputs. There are many possible approaches one could take to predict, we tried two simple approaches based on partial computation of dot products. Since the sign of a pre-activation input is determined by the sum of positive and negative partial products, we anticipate that the sum of an appropriately sampled subset of these terms would have the same sign as the full dot product. The first approach is selecting the top-k weights by magnitude, because the products with the larger magnitude weights are more likely to flip the sign. The second approach is random selection. We measure the precision and recall of the predictor at correctly predicting that a pre-activation input is negative.

Table 2: Sampling-based predictor results on DeiT-Tiny Touvron et al. (2021)

| SAMPLING RATE | 20% | | 10% | | 5% | | 1% | |
|---|---|---|---|---|---|---|---|---|
| | TOPK | RANDOM | TOPK | RANDOM | TOPK | RANDOM | TOPK | RANDOM |
| PRECISION | 0.94 | 0.93 | 0.93 | 0.94 | 0.93 | 0.94 | 0.94 | 0.95 |
| RECALL | 0.94 | 0.89 | 0.92 | 0.88 | 0.90 | 0.87 | 0.88 | 0.87 |

Table 2 shows that the precision and recall is high ($> 90\%$) for top-k sampling, while random sampling is close behind. Thus, we choose random sampling due to the practical implementation benefit that random sampling can be implemented by simple first-k selection of partial product terms. That is, for activations $X^{m \times n}$ and weights $W^{n \times p}$, we perform the operation $X^{m \times k} W^{k \times p}$, where $X^{m \times k}$ and $W^{k \times p}$ are the first $k$ columns of the activations and first $k$ rows of the weights, respectively. We call this the first-k sampling predictor (FKSP).

### 4.4.2 FKSP-MEDIATED CHANNEL SELECTION

To address challenge (2), our goal is to make it easier to separate computation that can be quantized from computation that requires high precision. We observed that some output channels exhibit a high proportion of negative pre-activation inputs, so we separate computation by channel. Note that this is the same dimension on which some weight quantization works like AWQ Lin et al. (2024) adapt their scaling factors, but instead of quantizing weights, we are quantizing the computation. First, we use the the FKSP to predict which pre-activation inputs are negative, then we compute the ratio of negative values in each channel. All pre-activation inputs in a channel with a negative ratio at or above a threshold are quantized, while channels with a negative ratio below the threshold are computed at full precision.

### 4.4.3 CACHED CHANNEL SELECTION

Empirically, we observed high correlation between channels selected in one batch to the next of around 90%. Thus, we use the first batch of images for classification tasks or of queries for LLMs as calibration data to select the channels to quantize and use the same quantization-full precision channel split for the rest of the batches. Section **??** investigates the difference between cached and dynamic FKSP-mediated channel selection, and finds little difference. While the performance of cached channel selection is slightly worse, it benefits greatly from not having to compute FKSP on every batch.

### 4.4.4 MIXED PRECISION INFERENCE

To achieve the goal in challenge (3), we design a framework to leverage INT4 and INT8 computation support on Turing GPUs to perform channel-wise mixed precision inference. First, we permute

Table 3: AFIQ for DeiTs shows smaller accuracy loss for the same proportion of computation quantized on larger models compared to small models.

| TOP-1 ACCURACY (%) ↑ / PROPORTION QUANTIZED (%) | TINY | SMALL | BASE |
|---|---|---|---|
| BASELINE | 72.1 / 0 | 79.8 / 0 | 81.8 / 0 |
| THRESHOLD = 0.01 | 71.9 / 11.0 | 79.3 / 17.1 | 81.1 / 41.1 |
| THRESHOLD = 0.05 | 71.4 / 22.0 | 79.0 / 28.5 | 81.0 / 54.7 |
| THRESHOLD = 0.1 | 71.3 / 28.9 | 78.8 / 36.6 | 80.8 / 60.7 |
| THRESHOLD = 0.15 | 71.0 / 33.7 | 78.7 / 42.5 | 80.7 / 64.4 |
| THRESHOLD = 0.2 | 70.9 / 37.2 | 78.5 / 47.1 | 80.6 / 67.3 |

channels so quantized and full precision channels are separated in their own contiguous groups. Permutation is done offline, so that runtime is not impacted, so sequentially swapping full precision channels with quantized channels with a higher index suffices. Similar permutation schemes have been proposed for sparsity and quantization Pool & Yu (2021); Zhang et al. (2024); Liu et al. (2025) to create multiple groups for adaptive scaling or for scheduling compute. In our case, we only require two contiguous groups for quantized and full precision computations, greatly simplifying the method. After the permuted FC1 and the activation function, the inputs to FC2 would become permuted, so FC2 weights need to be permuted along the input channel dimension to reverse the permutation. Thus, the activations following FC2 retain their original ordering.

At runtime, computing FC1 involves launching one INT4/INT8 kernel and one FP16 kernel. The INT4/INT8 kernel performs $Q(X)Q(W_{:,0:c}^T) + Q(\mathbf{b})$, where $Q$ is the quantization function to INT4/INT8 and $c$ is the number of quantized channels. Note that for weight quantized models, $W^T$ is already in INT4. The FP16 kernel performs $XD(W_{:,c+1:}^T) + \mathbf{b}$, where $D$ is the dequantization function to FP16. NVIDIA GPUs have API support for on-the-fly quantization and dequantization.

## 5 EVALUATION

We evaluate AFIQ by replacing FC1 (for non-SwiGLU models) or the gate projection layers (for SwiGLU models) with operations that simulate the functionality of AFIQ. The per channel quantization simulation is adapted from Xiao et al. (2023). For vision transformers, the model quality is measured by the top-1 accuracy (higher is better) on ImageNet-1K Deng et al. (2009). For LLMs, the model quality is measured by the perplexity (lower is better) on wikitext-2 Merity et al. (2016). We measure the speed up from mixed precision inference by implementing a mixed precision CUDA kernel and a full precision CUDA kernel using the cutlass framework [1]. For the First-k Sampling Predictor (§4.4.1), we sample the first 20% of the weights. The number of selected channels for quantization can be tuned based on the threshold on the proportion of negative predictions. We sweep this threshold to collect our results. Additional evaluations on channel selection stability can be found in Appendices F.

We evaluate the following models:

- Full precision ViTs: DeiT Touvron et al. (2021) and Swin-Transformer Liu et al. (2021).
- AWQ weight quantized LLMs: OPT-{125m,2.7B} Zhang et al. (2022), Llama-7B Touvron et al. (2023a), Llama2-7B Touvron et al. (2023b), Llama3-7B Grattafiori et al. (2024), and Qwen2.5-7B Team (2024).

### 5.1 MODEL QUALITY

Tables 3 and 4 show the accuracy and proportion of channels quantized of DeiT and Swin-Transformer models, respectively, as we sweep the threshold for quantizing to W4A4. Proportion of channels quantized directly correlates with the potential for speed up. DeiT-Tiny models suffer more accuracy loss for the same proportion of channels quantized when compared to DeiT-Base models. Curiously,

---

[1]https://github.com/NVIDIA/cutlass

Table 4: AFIQ on Swin-Transformers shows accuracy increases with increased proportion of computation quantized, with better accuracy-quantization tradeoffs for larger models.

| TOP-1 ACCURACY (%) ↑ / PROPORTION QUANTIZED (%) | SWIN-TRANSFORMER | | | SWIN-TRANSFORMER-V2 | | |
|---|---|---|---|---|---|---|
| | TINY | SMALL | BASE | TINY | SMALL | BASE |
| BASELINE | 81.2 / 0 | 83.2 / 0 | 83.4 / 0 | 82.8 / 0 | 83.7 / 0 | 84.6 / 0 |
| THRESHOLD = 0.01 | 81.1 / 19.4 | 83.2 / 14.1 | 83.4 / 14.4 | 83.9 / 41.8 | 85.4 / 35.7 | 85.3 / 24.0 |
| THRESHOLD = 0.05 | 81.0 / 36.4 | 83.1 / 30.1 | 83.4 / 29.8 | 83.5 / 63.2 | 86.0 / 54.4 | 85.9 / 35.5 |
| THRESHOLD = 0.1 | 80.9 / 46.1 | 83.1 / 40.7 | 83.4 / 39.0 | 82.5 / 73.3 | 85.4 / 64.7 | 86.1 / 43.0 |
| THRESHOLD = 0.15 | 80.9 / 52.2 | 83.0 / 47.4 | 83.4 / 45.2 | 81.1 / 78.3 | 84.2 / 71.2 | 86.0 / 48.9 |
| THRESHOLD = 0.2 | 80.8 / 56.6 | 83.1 / 52.3 | 83.3 / 49.9 | 80.4 / 81.4 | 83.2 / 76.0 | 85.9 / 53.8 |

Table 5: Since W8A8 computation is applied to LLMs, the perplexity is hardly affected by quantization.

| PERPLEXITY ↓ / PROPORTION QUANTIZED (%) | OPT | | | LLAMA | | | QWEN |
|---|---|---|---|---|---|---|---|
| | 125M | 1.3B | 6.7B | 7B | 2-7B | 3-8B | 2.5-7B |
| AWQ | 29.1 / 0 | 14.9 / 0 | 11.0 / 0 | 5.8 / 0 | 5.6 / 0 | 6.5 / 0 | 7.1 / 0 |
| THRESHOLD = 0.01 | 29.1 / 8.2 | 14.9 / 28.7 | 11.0 / 49.9 | 5.8 / 0.3 | 5.6 / 0.2 | 6.5 / 1.7 | 7.1 / 22.9 |
| THRESHOLD = 0.05 | 29.3 / 32.7 | 15.0 / 76.0 | 11.0 / 96.1 | 5.8 / 20.9 | 5.6 / 19.7 | 6.5 / 51.0 | 7.1 / 76.7 |
| THRESHOLD = 0.1 | 29.5 / 58.9 | 15.0 / 94.3 | 11.0 / 99.9 | 5.8 / 81.2 | 5.6 / 81.7 | 6.5 / 97.2 | - |
| THRESHOLD = 0.15 | 29.7 / 75.2 | 15.0 / 98.8 | - | 5.8 / 99.2 | 5.6 / 99.3 | - | - |
| THRESHOLD = 0.2 | 29.8 / 85.0 | 15.0 / 99.8 | - | - | - | - | - |

Swin-Transformer accuracy actually increases with quantization for the Swin-Transformer-V2 models. Table 5 shows the perplexity for LLMs quantized to W8A8 since perplexity exploded with W4A4 computation. The empty entries indicate that applying the threshold would lead to a fully quantized model.

## 5.2 SPEED UP FROM MIXED PRECISION INFERENCE

Using the linear layer dimensions from the evaluated vision transformers, we executed the INT4-FP16 mixed precision and FP16 kernels, and measured the speed up of the former over the latter. The proportion quantized was set to the proportion that was corresponds to the quantized model accuracy being within 1% of baseline. Fig. 3 shows the speed up. While some models experience slight slowdowns due to the GPU matrix size effects, there is an overall 17% speed up from using the mixed precision kernel.

## 6 DISCUSSION

### 6.1 HARDWARE SUPPORT FOR LOW-PRECISION FORMATS

While this work evaluated INT4/INT8 quantized computation, other low-precision formats have begun to enjoy hardware support. While we evaluated the speed up of mixed precision INT4/INT8-FP16 computation on a Turing architecture GPU building on weight quantization works that use INT4 (AWQ and GPTQ), the latest NVIDIA Blackwell GPU family supports FP4 computation NVIDIA (2025a) and removes support for INT4 computation. Additionally, FP4 weight quantization models have begun emerging NVIDIA (2025b). We prioritized evaluating AFIQ with AWQ and GPTQ because of they are well-established techniques, but we have no doubts AFIQ would be effective for mixed FP4-FP16 quantized computation. FP4 computation would introduce quantization error that AFIQ would mitigate via selective channel-wise quantization. Future GPUs could bring back support for integer formats because at narrower bitwidths, integer computation can be efficiently implemented with look up tables Mo et al. (2025).

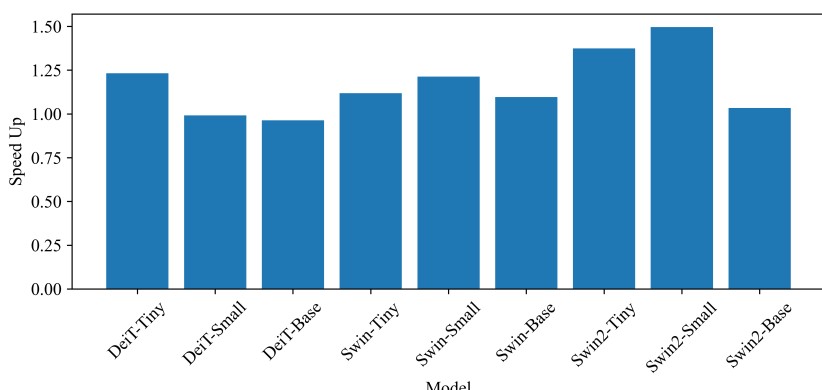

Figure 3: Mixed precision kernel speed up compared to the FP16 kernel shows speed up for most models, with an average speed up of 17% speed up

## 6.2 Generalizing to Other Nonlinearities

This work focusses on quantizing linear layer computation due to the computational bottleneck of that layer. However, the fundamental concept of selective quantization based on the gradient of the next operation can equally apply to nonlinearities other than activation functions. For example, the Softmax nonlinearity following the attention probability computation (Fig. 2) has an input-dependent gradient. Analysis of this gradient is elaborated in Appendix E. Designing a predictor to estimate the gradient such that we could apply selective quantization of the attention probability computation is left to future work.

## 6.3 Limitations and Future Work

A limitation of AFIQ is that it cannot achieve full W4A4 or W8A8 quantized computation as other methods like RPTQ Yuan et al. (2023) or SmoothQuant Xiao et al. (2023) propose. The scope of this study is also limited to uniform per-channel quantization with integer low-precision formats. Integrating better quantization schemes like reordered group quantization Yuan et al. (2023) or using normal float Dettmers et al. (2023) could improve model quality at lower precision.

## 7 Conclusion

We address the computational bottleneck in linear layers of vision transformers and large language models with selective quantization of the computation. Based on the observation that for negative inputs, activation functions have a small gradient magnitude, we propose Activation Function Informed Quantization (AFIQ). Using a low-overhead sampling-based predictor to exploit this observation, AFIQ selects output channels whose dot products should be computed in INT4 or INT8 instead of FP16. This channel selection can be selected once based on a single calibration sample and reused to make the overhead negligible. Overall, AFIQ can speed up linear layer computation by 17% without significant effects on model quality.

## 8 Reproducibility Statement

Open source code to reproduce this work is provided at this Github link[2]. This reinforces the details in the method section (§4) meant to aid reproducibility.

---

[2]https://github.com/AFIQ-submission/afiq

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

## A   COMPUTE BOTTLENECK

We present a simple example to show how quantizing weights without quantizing computation can lead to a compute bottleneck. Consider a DeiT-Tiny Touvron et al. (2021) model running on the latest NVIDIA Blackwell GPU. The GB200 chip is capable of 5 petaFLOPS at FP16 and has 16 TB/s of memory bandwidth NVIDIA (2025a), resulting in 312.5 FLOPs/byte of arithmetic intensity. Based on layer dimensions, DeiT-Tiny with 4-bit weights and FP16 computation would have an arithmetic intensity in its linear layers of 380 FLOPs/byte, resulting in a compute bottleneck. In other words, for DeiT-Tiny model inference to consume all available memory bandwidth, the GPU would need to provide $1.22\times$ more FLOPs of compute in the same amount of time.

## B   ACTIVATION FUNCTIONS

### B.1   GELU

As GELU is the most popular, we will analyze it in more detail. We will use the following approximation of GELU Hendrycks & Gimpel (2016):

$$\text{GELU}(x) = 0.5x\left[1 + \tanh\left(\sqrt{\frac{2}{\pi}}\left(x + 0.044715x^3\right)\right)\right] \tag{6}$$

so its gradient is

$$\frac{\mathrm{d\,GELU}}{\mathrm{d}x} = 0.5 \left[ 1 + \tanh\left( \sqrt{\frac{2}{\pi}} \left( 0.044715x^3 + x \right) \right) \right]$$
$$+ 0.5 \sqrt{\frac{2}{\pi}} \left( x + 3 \cdot 0.044715x^3 \right) \mathrm{sech}^2 \left( \sqrt{\frac{2}{\pi}} \left( x + 0.044715x^3 \right) \right) \tag{7}$$

Plots of GELU's gradient (Fig. 1) show that the gradient is small for most of the negative component of the domain and tops out at 0.5 at $x = 0$. In the domain of [-8, 0] (matching the distribution of output values in Fig. 4), the gradient of GELU has an average value of 0.0425.

## B.2 ReLU

As the output of ReLU is 0 for all negative inputs, it is not even necessary to compute these negative preactivations. Knowing what computation would result in negative preactivations such that they can be avoided is challenging, but it has been addressed in prior work, namely SnaPEA Akhlaghi et al. (2018). Discussion of why SnaPEA does not work for other activation functions is given in Appendix C.

## B.3 SwiGLU

SwiGLU Shazeer (2020) is a special activation function in that it combines the non-linear function, SiLU, with matrix multiplication between the input ($x$) and three weight matrices ($W_1$, $W_2$, $W_3$).

$$\mathrm{SwiGLU}(x) = W_2 * \mathrm{SiLU}(W_1(x) * W_3(x)) \tag{8}$$

where SiLU Hendrycks & Gimpel (2016) is given by $x\sigma(x)$.

Plots of SiLU's gradient (Fig. 1) show that the gradient is small for most of the negative component of the domain and tops out at 0.5 at $x = 0$. In the domain of [-8, 0], the gradient of SiLU has an average value of 0.0693.

## C ReLU-based Approaches

In this section, we discuss in more depth the problems with applying ReLU-based prior approaches to transformers, as well quantitative results of trying to do so.

We consider SnaPEA Akhlaghi et al. (2018), a representative early termination technique. It exploits the following two characteristics of ReLU in CNNs: (1) the ReLU function output zero for any negative input, so if one can predict which pre-activations will be negative, then one can skip computing the value of that negative pre-activation; and (2) CNNs have repeating sequences of convolution followed by ReLU, which means that the input to most convolution layers are strictly non-negative. SnaPEA Akhlaghi et al. (2018) orders weights from most positive to most negative, then computes and accumulates partial products serially until the partial product becomes negative. From observation (2), the partial products must decrease monotonically as they are computed, and from observation (1), it outputs zero once the partial product becomes negative.

The two assumptions underlying ReLU-based early negative termination both fall apart in transformers. Observation (1) fails for non-RELU activation functions, because it is not enough to make a binary (output = 0 or something else) prediction when negative pre-activation values map to non-zero negative post-activation values. We swept a range of predicted single values and none achieve the performance of AFIQ, despite the benefit of an unrealistic oracle predictor. Even if the transformer used ReLU like OPT Zhang et al. (2022) does, this method would run into problems with observation (2) not being true. As shown in Fig. 2, the inputs into the pre-activation computation are not the output of ReLU and thus are not strictly non-negative. Fig. 4 shows that these inputs follow a normal distribution, clearly violating that asssumption underlying these methods.

## D  DISTRIBUTION OF PRE-ACTIVATION INPUTS

Fig. 4 shows histograms of the distribution of FC1 activations, FC1 weights, FC1 biases, and pre-activation inputs for all layers of a SwinTransformer Liu et al. (2021) on ImageNet-1K Deng et al. (2009). The input values follow a normal distribution and this is expected because the prior layer is a normalization layer, as shown in Fig. 2. While the trained weight values also appear to follow a normal distribution, the addition of negatively-skewed bias terms causes the distribution of pre-activation inputs to skew negative.

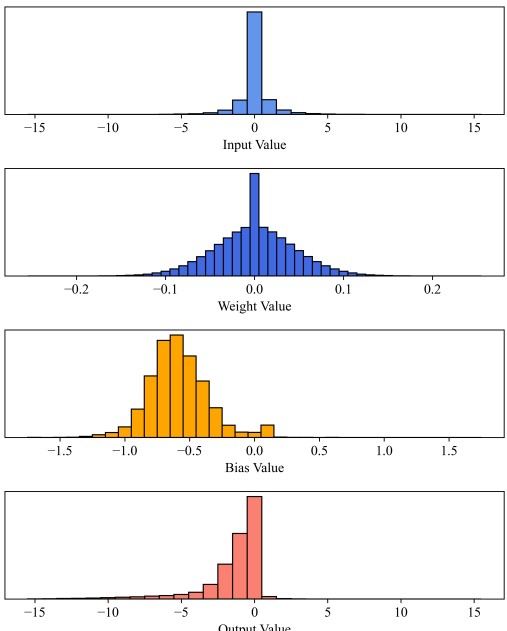

Figure 4: Relative frequency of input, weight, bias, and output values of the linear layers of a SwinTransformer. While the input and weight values are normally distributed, the bias values skew negative, causing the output to also skew negative.

## E  SOFTMAX GRADIENT

In this section, we explore another nonlinearity, Softmax, in the context of successfully exploiting the characteristics of the activation function nonlinearity. Softmax is used to calculate the attention weight from the attention probabilities. Since the calculation of attention probabilities varies from model to model, we consider a generic linear transformation. Consider the computation of an attention weight slice, as softmax is typically performed on all slices along a particular dimension. The attention weight is given by

$$w_a = \sigma(\mathrm{F_p}(X)) \tag{9}$$

where $w_a$ is the attention weight slice, $\sigma$ is the softmax function, $\mathrm{F_p}$ is the linear transformation $R^{N \times d} \to R^N$ that produces the attention probability slice and $X$ is the input matrix containing weight vectors, prior layer activation vectors, masks, etc. For convenience, define the attention probability $z = \mathrm{F_p}(X)$, so $w_a = \sigma(z)$.

The $i^{\mathrm{th}}$ element of the attention weight slice is given by

$$\sigma(z_i) = \frac{e^{z_i}}{\sum_{k=1}^{K} e^{z_k}}, \tag{10}$$

Table 6: Channel Selection Stability (%) for DeiT

| TINY | SMALL | BASE |
|------|-------|------|
| 97.3 | 97.3  | 94.6 |

Table 7: Channel Selection Stability (%) for Swin-Transformers

| SWIN-TRANSFORMER | | | SWIN-TRANSFORMER-V2 | | |
|------|-------|------|------|-------|------|
| TINY | SMALL | BASE | TINY | SMALL | BASE |
| 95.9 | 96.6  | 95.0 | 88.7 | 90.6  | 94.5 |

where $K$ is the size of slice $z$. Its gradient is

$$\frac{\partial\sigma(z_i)}{\partial z_j} = \sigma(z_i)(1\{i=j\} - \sigma(z_j)) = \frac{e^{z_i}}{\sum_{k=1}^{K} e^{z_k}}\left(1\{i=j\} - \frac{e^{z_j}}{\sum_{k=1}^{K} e^{z_k}}\right) \tag{11}$$

where $1\{i=j\}$ is 1 if $i=j$ and 0 otherwise Kuribel (2021).

The gradient equation appears to frustrate our efforts to quantize any of the inputs to compute the attention weight, since computing the gradient of the softmax requires computing the softmax itself. This is once again a circular dependency. However, we can still gain insight into when the magnitude of the gradient would be larger or smaller. Consider that the sum of exponentials $\sum_{k=1}^{K} e^{z_k}$ is in the denominator and $e^{z_i}$ is in the numerator, so

$$\frac{\partial\sigma(z_i)}{\partial z_j} \propto \frac{1}{\sum_{k=1}^{K} e^{z_k}} \quad \text{and} \quad \frac{\partial\sigma(z_i)}{\partial z_j} \propto e^{z_i} \tag{12}$$

## F  CHANNEL SELECTION STABILITY

Tables 6,7, and 8 show channel selection stability, which is the correlation between the channel selection for every batch and the channel selection for the first batch. The threshold parameter is set at 0.01. Note that the channel selection for the first batch is cached and reused for all subsequent batches. The channel selection stability is generally over 90%.

## G  ORACLE STUDY

In this section, we analyze the effect of quantizing all dot products that result in negative pre-activation inputs. While this is not practical to implement, it does directly test the key hypothesis of this work around which computations to quantize. Tables 9,10, and 11 show the oracle performance along with the proportion of dot products quantized. While model quality is degraded, it is usually not catastrophic. Notice that the quantization rates are between 69.1% and 97.0%, demonstrating the large number of negative pre-activation inputs in transformers.

Table 8: Channel Selection Stability (%) for LLMs

| OPT | | | LLAMA | | | QWEN |
|------|------|------|------|------|------|--------|
| 125M | 1.3B | 6.7B | 7B | 2-7B | 3-8B | 2.5-7B |
| 98.4 | 90.6 | 83.7 | 99.9 | 99.9 | 99.1 | 91.9 |

Table 9: Oracle study for DeiT

| TOP-1 ACCURACY (%) ↑ / PROPORTION QUANTIZED (%) | TINY | SMALL | BASE |
|---|---|---|---|
| BASELINE | 72.1 / 0 | 79.8 / 0 | 81.8 / 0 |
| ORACLE | 70.7 / 84.8 | 78.6 / 92.0 | 80.8 / 94.5 |

Table 10: Oracle study for Swin-Transformer models

| TOP-1 ACCURACY (%) ↑ / PROPORTION QUANTIZED (%) | SWIN-TRANSFORMER-V2 | | |
|---|---|---|---|
| | TINY | SMALL | BASE |
| BASELINE | 82.8 / 0 | 83.7 / 0 | 84.6 / 0 |
| ORACLE | 82.3 / 80.1 | 83.4 / 85.4 | 84.6 / 88.0 |

Table 11: Oracle study for LLMs.

| PERPLEXITY ↓ / PROPORTION QUANTIZED (%) | OPT | | | LLAMA | | | QWEN |
|---|---|---|---|---|---|---|---|
| | 125M | 1.3B | 6.7B | 7B | 2-7B | 3-8B | 2.5-7B |
| AWQ | 29.1 / 0 | 14.9 / 0 | 11.0 / 0 | 5.8 / 0 | 5.6 / 0 | 6.5 / 0 | 7.1 / 0 |
| ORACLE | 8547.9 / 92.9 | 154.5 / 97.0 | 8718.7 / 96.6 | 5.8 / 69.1 | 5.6 / 70.2 | 6.5 / 76.7 | 7.1 / 86.7 |

