# OpenReview forum: "Activation Function Informed Quantization"
_ICLR.cc/2026/Conference — Submitted to ICLR 2026_

### Official Review · Reviewer_BA3p · 2025-10-26

**Soundness:** 3
**Presentation:** 3
**Contribution:** 2
**Rating:** 4
**Confidence:** 4

**Summary:**

The paper first reports that negative values are less sensitive to quantization and proposes (1) sampling values at the channel granularity, (2) applying lower precision to negative value-rich channels, and (3) a mixed precision GPU kernel to perform INT8 (for negative) and FP16 computation (others) thereby obtaining speedup due to FP16 to INT8 conversion for negative rich channels.

**Strengths:**

The idea of applying low precision to negative values looks simple but new and solid based on the observations.
The effects of average 17% latency reduction looks quite good.

**Weaknesses:**

It would be fantastic if FP8 based comparison were provided for LLM.
Currently, FP8 is the main stream in LLM.
Thus, FP8 (baseline) vs FP4-FP8 (the authors') comparison is needed to claim the utility.
Especially, NVFP4-FP8 (based on the authors' idea) would be promising since NVFP4 offers fine-grained and 2-level scaling and, therefore, is favorable to the proposed idea.

More detailed analysis on latency reduction would be helpful.
For instance, ViTs benefit more from the proposed idea than LLMs
possibly due to the dominance of FFN on ViT latency.

**Questions:**

In LLMs, what would the FP8 (baseline) vs FP4-FP8 (proposed) comparison be like?
Why does ViT show more latency reduction than LLM?

---

### Official Review · Reviewer_JZc6 · 2025-10-29

**Soundness:** 2
**Presentation:** 2
**Contribution:** 2
**Rating:** 2
**Confidence:** 4

**Summary:**

Based on the observation that the gradients of transformer activation functions (ReLU, GELU, SiLU) are small when inputs are
negative, the authors introduce AFIQ, a weight-activation PTQ method, which samples dot-product partial products on a single calibration example to determine which channels to quantize for all future model inference.

**Strengths:**

1. The authors deal with various models ranging from ViTs to LLMs with different quantization schemes such as W4A4 and W8A8.
2. They show the acceleration effect of the proposed method for ViTs.

**Weaknesses:**

1. The authors mentioned in Abstract that they offer a novel approach for W4A4 and W8A8 (i.e., weight-activation quantization), but in Table 5, they only compared the proposed method with AWQ, which is designed for weight-only quantization. In order to demonstrate the efficacy of the proposed method for W8A8 in Table 5, they should reference important W8A8 works such as SmoothQuant [1] and FlexRound [2], but they do not compare the proposed method with either SmoothQuant or FlexRound in Table 5.
2. It is already well known that LLM W8A8 quantization can perform comparable to FP baselines. In other words, W8A8 is easy-to-quantize for LLMs. Accordingly, to show the effectiveness of the proposed method as a LLM weight-activation quantization technique, they should conduct experiments with W4A4, but there is no experiments about W4A4 for LLMs.
3. In Abstract, the authors highlighted a mixed-precision linear layer kernel in CUDA that can be 17% faster than FP16 baseline, but it is only shown in Figure 3 for VITs, not LLMs. As a result, it is doubtful whether the proposed kernel can be also effective for LLMs.

[1] SmoothQuant: Accurate and Efficient Post-Training Quantization for Large Language Models, ICML 2023

[2] FlexRound: Learnable Rounding based on Element-wise Division for Post-Training Quantization, ICML 2023

**Questions:**

In line 316-317, there is a typo, "Section ??". Which section does it originally indicate?

---

### Official Review · Reviewer_Zk6z · 2025-10-30

**Soundness:** 2
**Presentation:** 2
**Contribution:** 2
**Rating:** 2
**Confidence:** 4

**Summary:**

This paper proposes AFIQ, a PTQ method for quantizing transformer-based models. The key insight is that activation functions (ReLU, GELU, SiLU) exhibit small gradients for negative pre-activation inputs, suggesting that quantization errors in those regions have limited downstream impact. Therefore, AFIQ selectively applies INT4 or INT8 quantization to those channels whose pre-activation inputs are mostly negative.

**Strengths:**

The paper presents experiments on both vision transformers (e.g., DeiT, Swin-Transformer) and large language models (e.g., OPT, LLaMA, Qwen), demonstrating the generality of the proposed quantization method across distinct architectures and modalities. This strengthens the claim that Activation Function Informed Quantization (AFIQ) is broadly applicable rather than tailored to a specific model family. The authors also report consistent speedup trends across these diverse tasks, which adds credibility to the method’s robustness in real-world inference scenarios.

**Weaknesses:**

1. Writing Quality and Presentation Issues: The paper suffers from poor writing quality and formatting inconsistencies, which detract from its readability and professionalism.

2. Inadequate Baseline Comparison and Outdated References: While the authors compare AFIQ against AWQ and a few older quantization methods, these are no longer representative of the state of the art in PTQ. More recent techniques such as QUIP#, SPINQuant, OMNIQuant have demonstrated superior performance in both accuracy retention and quantization flexibility. The absence of comparisons with these contemporary PTQ methods raises concerns about whether AFIQ’s improvements (e.g., 17% speedup) remain competitive under fair, modern benchmarks. Including stronger baselines or at least discussing how AFIQ conceptually differs from these newer methods would provide necessary context and improve the paper’s credibility.

3. Insufficient Theoretical Justification for Gradient-Based Motivation: The central intuition of AFIQ—that activation gradients can serve as a proxy for quantization sensitivity or output importance—is intriguing but not rigorously justified. The paper provides empirical evidence that gradients are small for negative pre-activation values, but it does not convincingly explain why or how this property correlates with downstream accuracy robustness. There is no formal analysis (e.g., sensitivity study, Taylor expansion argument, or gradient-based importance correlation) that links gradient magnitude to the expected error propagation in transformer outputs. Without a clearer theoretical or empirical justification, the connection between the observed gradient behavior and quantization errors feels somewhat heuristic. Strengthening this argument would make the underlying insight more compelling and generalizable.

**Questions:**

See above

---

### Official Review · Reviewer_5f2w · 2025-10-31

**Soundness:** 2
**Presentation:** 3
**Contribution:** 2
**Rating:** 2
**Confidence:** 3

**Summary:**

The paper introduces Activation Function Informed Quantization, a quantization technique that quantizes linear layers (w4a4 or w8a8) with selective channel wise quantization.
The paper proposes to predict the sign of pre-activation values using a lightweight predictor, and quantizing the channels that are likely to be negative into int4/8 while keeping rest in higher precision. This approach requires a single calibration example and the efficacy of the algorithm is shown via a custom CUDA kernel with 17% performance benefits.

**Strengths:**

1. The results demonstrate 17% performance benefits without any retraining.
2. Minimal accuracy loss via calibration through a single representative example.
3. The paper implements a mixed precision CUDA kernel.

**Weaknesses:**

1. Efficacy of the algorithm depends on the sign prediction algorithm and may not be stable across architectures.
2. Requires complex caching logic to implemented in inference for adding the support for this algorithm.
3. This approach needs to be validated across broader family of LLMs, limited empirical proof.
4. Calibration data might degrade accuracy when LLMs are trained to solve 100s of different and varied tasks.

**Questions:**

1. How does accuracy of predictor vary based on different activation functions?
2. Do you have more performance data of accuracy degradation when domain shifts compared to the calibration example?

---

### Author Response · Authors · 2025-11-26
**Thank you for your comments**

Thank you to the reviewers for your thoughtful comments. I humbly suggest that the gap between your expectations and the state of this paper are too large to be closed by work during the rebuttal period. I intend to work to address the issues raised before resubmitting to another venue with additional work to do the following:
- build on prior work that already achieves 4bit quantization on LLMs to achieve sub-4bit quantization
- validation across more LLMs, including additional tasks
- show calibration can generalize from one dataset to another and one task to another
- ablation studies and sensitivity studies
- evaluation with two-level scaling based quantization formats
- explain that while the ReLU and GELU activation functions are evaluated in this work, other activation functions can also work

---

### Meta-Review · Area_Chair_8TCK · 2026-01-03

**Summary:**

Multiple reviewers raised concerns about (i) limited novelty relative to recent PTQ methods (Zk6z, JZc6), (ii) insufficient comparison against more recent baselines (Zk6z, JZc6), and (iii) incomplete evaluation for LLM settings (5f2w, JZc6, BA3p).  While the idea shows moderate speedups, the empirical evidence on limited settings is not strong enough to make it stand out.

**Reviewer Concerns:**

No individual rebuttal was provided to each review. It seems the authors acknowledged the limitations raised by the reviewers and decided to work on improving those areas for another submission.

**Reviewer Scores:**

Since no individual rebuttal is provided, I expect no change in reviewer's positions.

---

### Decision · Program_Chairs · 2026-01-26

Reject